# Semi-supervised Learning with Contrastive and Topology Losses for Catheter Segmentation and Misplacement Prediction

**Tianyu Hwang**[1]                                                    tyhwang@ncts.ntu.edu.tw
**Chih-Hung Wang**[2]                                           ogenkidesga@gmail.com
**Holger R. Roth**[3]                                                  hroth@nvidia.com
**Dong Yang**[3]                                                        dongy@nvidia.com
**Can Zhao**[3]                                                          canz@nvidia.com
**Chien-Hua Huang**[2]                                        chhuang5940@ntu.edu.tw
**Weichung Wang**[1]                                            wwang@ntu.edu.tw

[1] *National Taiwan University, Taiwan*

[2] *National Taiwan University Hospital, Taiwan*

[3] *NVIDIA, Bethesda, Santa Clara, USA*

**Editors:** Accepted for publication at MIDL 2023

## Abstract

Chest X-ray images are often used to determine the proper placement of catheters, as incorrect placement can lead to severe complications. With the advent of deep learning, computer-aided detection methods have been developed to assist radiologists in identifying catheter misplacement by detecting and highlighting the catheter's path. However, obtaining large, pixel-wise labeled datasets can be challenging due to the labor-intensive nature of annotation. To address this issue, we proposed a novel semi-supervised learning method that combines contrastive loss and topology loss. This method takes advantage of the known topological properties of catheters and does not require extensive labeling. We collected 7,378 chest X-ray images from the National Taiwan University Hospital, which were labeled for misplacement of nasogastric and endotracheal tube catheters, and included pixel-wise annotation. Moreover, the CLiP dataset was used as an unlabeled dataset for semi-supervised learning. We used a hybrid U-Net architecture with an added classification head to perform simultaneous segmentation of the catheter and misplacement classification. Our model achieved average AUC of 0.977 for classification and average Dice score of 0.614 for segmentation.

**Keywords:** Semi-supervised Learning, Contrastive Learning, Topology Loss, Catheter Misplacement, Catheter Segmentation

## 1. Introduction

Chest X-ray (CXR) is widely used to evaluate the proper placement of catheters, as incorrect placement can lead to serious complications. Misplacement of an endotracheal tube (ETT) can result in conditions such as hypoxemia, pneumothorax, and pneumonia (Lakhani et al., 2021), while a misplaced nasogastric tube (NGT) can cause respiratory failure and even death (Singh et al., 2019). Studies have shown that the estimated misplacement rate of ETT and NGT catheters can be as high as 28% and 15%, respectively (Yi et al., 2020). In particular, X-rays that were captured using a portable X-ray device have low contrast and

high noise, making it challenging for clinicians to visually detect the position of catheter tips. A clinical trial (Torsy et al., 2018) demonstrated that the image quality was insufficient to acquire conclusive visibility of NGT position in 16.9% of portable CXRs. Well-trained radiologists are essential to confirm catheter position on CXRs, but they may not always be immediately available.

This is why we aimed to develop a deep learning-based computer-aided detection (CAD) system to facilitate the localization of the catheter and detect its malposition on CXRs. With the advancement of computer vision and deep learning, CAD methods have been developed to assist radiologists in quickly and accurately detecting catheter misplacement (Aryal and Yahyasoltani, 2021; Elaanba et al., 2021; Lakhani et al., 2021; Lakhani, 2017; Singh et al., 2019). However, for CAD to be clinically useful, it is important to also consider model explainability. This has led to the development of semantic segmentation methods for catheter placement on CXRs, which can provide detailed information about the location and shape of the catheter (Sullivan et al., 2020; Gherardini et al., 2020).

However, obtaining large datasets with pixel-wise labels is often difficult as it requires a lot of resources and expert knowledge to perform the annotation of the images. Additionally, class imbalance is a common problem in catheter placement classification, as misplaced catheters are less common than correctly placed catheters. This makes it challenging for a model to accurately segment misplaced catheters and predict the absence or misplacement of catheters. Semi-supervised contrastive learning techniques have been used to address these issues by utilizing large unannotated datasets (Hu et al., 2021).

At the same time, realizing that the catheter is a physically connected object, we can use its expected topology as a prior to compute a topology loss. This loss does not require labeling as the catheter's topology is assumed to be the same across all images. Therefore, we propose a semi-supervised learning method that combines a contrastive loss with a topology loss to improve the performance of catheter placement classification and segmentation.

We developed an algorithm that simultaneously performs catheter segmentation and misplacement prediction using a U-Net architecture with an additional classification head. This model can predict the misplacement of the ETT or NGT catheters, in addition to segmenting the catheters and relevant anatomical landmarks. This allows our model to provide not only detailed information about the catheter location but also its malposition state, and hence make the model more useful in clinical practice and explainable.

Our contributions in this work can be summarized as follows:

- Developed a catheter segmentation and misplacement prediction deep learning model.
- Validated the proposed algorithm on an external testing set.
- Combined contrastive and topology losses for semi-supervised learning.
- Modified topology loss to add constraints on a range of topological features.

## 2. Methods

### 2.1. Model

The model we use is based on U-Net (Ronneberger et al., 2015), a popular fully convolutional neural network architecture for biomedical image segmentation. U-Net contains an

encoder that extracts features of different spatial resolutions, and uses skip connections to pass feature maps to a decoder that generates a segmentation mask through pixel-wise classification. The encoder in our model uses ResNeSt-50 (Zhang et al., 2020) as its backbone, which implements a channel-wise attention mechanism to capture cross-feature interactions and learn diverse representations. Additionally, we apply Spatial and Channel Squeeze and Excitation (scSE) (Roy et al., 2019) to the decoder blocks, which aggregates spatial information and feature information to enhance the segmentation performance. To perform both segmentation and misplacement classification, we add a feature projector and a classification head after the encoder, in addition to the U-Net architecture for segmentation. The feature projector is used to convert the features extracted by the encoder into a feature representation that can be used in the contrastive learning. For the architecture of the model, see Appendix A.

## 2.2. Contrastive Loss

Contrastive loss is a loss function used in self-supervised contrastive learning, which aims to learn visual representations from unlabeled images. The framework we use is the Sim-CLR framework proposed by (Chen et al., 2020). The approach is to train the model to differentiate between different views of the same data. This allows the model to learn representations of the data that can be used for downstream tasks such as segmentation and classification.

The implementation is as follows, for each input batch $B = \{x_1, x_2, ..., x_b\}$, where $x_i$ represents each input image, every image is performed augmentations twice to obtain $B' = \{x_1, x_2, ..., x_{2b}\}$. Let $\sigma(i)$ be the index of image that is derived from the same image as index $i$. Then denote the encoder as $E(\cdot)$ and the feature projector as $g(\cdot)$. The contrastive loss of a batch is defined as

$$\text{Contr}(B') = -\frac{1}{2b} \sum_{i=1}^{2b} \log \frac{\exp(\text{sim}(z_i, z_{\sigma(i)})/\tau)}{\sum_{k=1}^{2b} \mathbb{1}_{[k \neq i]} \exp(\text{sim}(z_i, z_k)/\tau)} \tag{1}$$

where $\tau$ is the temperature, $z_i = g(E(x_i))$ and $\text{sim}(u, v) = \frac{u^T v}{||u|| \cdot ||v||}$ which is the cosine similarity. The augmentations we are using is the composition of random contrast adjustment, Gaussian noise, dropout, random grid distortion and random affine transformation.

## 2.3. Topology Loss

The topology loss (Clough et al., 2020) is a segmentation loss that is based on persistent homology, a mathematical framework for the study of topological features of shapes. The topology loss does not require segmentation labels but instead uses topology priors, such as Betti numbers, to impose constraints on the segmentation output. The Betti numbers are topological invariants that describe topological properties such as the number of connected components and the number of holes in a shape. The loss function consists of two components, the level set and the barcode diagram. The level set is the collection of segmentation results at different probability thresholds, and the barcode diagram encodes information about the topological changes that occur as the threshold is varied.

Notice that when the threshold is below zero, the entire segmentation would be considered one connected component. In theory, this would lead to an infinitely long bar in the

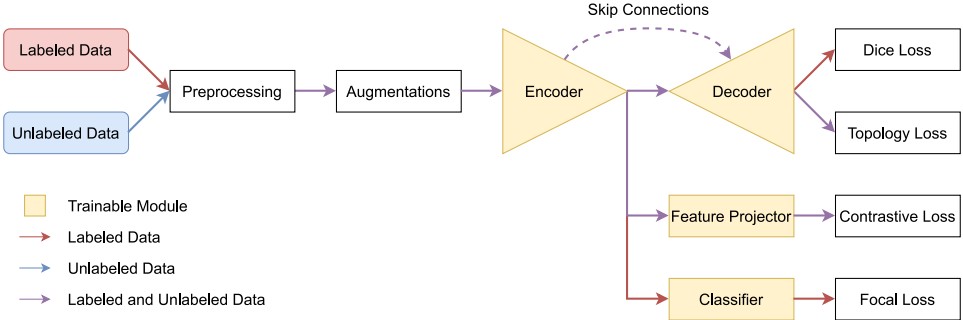

Figure 1: The complete semi-supervised training pipeline.

barcode diagram. Hence, the authors of the topology loss suggested that this bar can be cut off at a threshold of 0 (Clough et al., 2020). We implemented the topology loss such that the bar is cut off at zero. By doing this, we can impose the segmentation to be either zero or one connected component.

We also modified the loss such that the target can be applied on a range of acceptable Betti numbers, instead of an exact value. For example, in our case the catheter might or might not be present in the image, thus the number of connected components can be zero or one. To be able to use this loss, we can modify its implementation such that we only penalize when the output is beyond our target range. Thus, the topology loss of an segmentation output is defined as

$$\text{Topo}(\hat{y}) = \sum_{k} \left( \sum_{l=1}^{\beta_{k,a}} (1 - |B_{l,k}(\hat{y})|)^2 + \sum_{l=\beta_{k,b}+1}^{\infty} |B_{l,k}(\hat{y})|^2 \right) \tag{2}$$

where $|B_{l,k}(\hat{y})|$ is the length of $l$-th longest barcode of dimension $k$ of output $\hat{y}$. $\beta_{k,a}$ and $\beta_{k,b}$ are respectively the lower and upper bound of the acceptable Betti number of dimension $k$. Notice that the second term has infinitely many terms, but in practice there can only be a finite number of topology features in output of a finite image. In fact, most barcodes will be zero, hence we set the the maximum number of terms to be 20.

In our task, we apply the topology loss to the segmentation prediction of catheters, where it can be at most one connected component. We also did not impose a constraint on the number of holes present in the segmentation, since it is possible for catheter to curl and form loops. It is important to note that computing this topological loss is computationally expensive and is currently only implemented on CPU since it could not be easily parallelized on the GPU. To mitigate this issue, we downscale the output by max-pooling to a resolution of $16 \times 16$. The computation of the topology loss scales with the number of pixels, with the training time for each iteration at resolutions of $16 \times 16$, $64 \times 64$, and $256 \times 256$ being 1.0s, 1.5s, and 17.5s, respectively. Despite the decreased resolution, the overall structure of the segmentation prediction is still preserved.

## 2.4. Semi-supervised Learning

Figure 1 shows the semi-supervised training pipeline that we proposed. Noticing that both the contrastive loss and the topology loss do not require labels to be computed, we combine

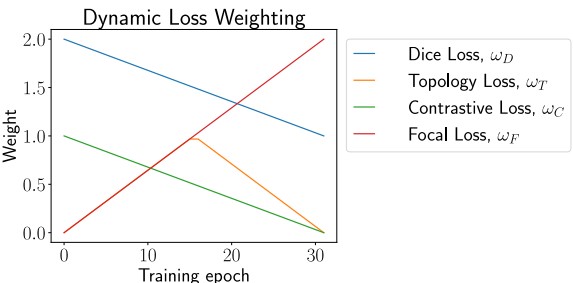

Figure 2: Weighting of the losses that changes throughout the training.

both method for semi-supervised learning. The total loss for a batch is defined as

$$\text{Total Loss} = \omega_D \text{Dice}_l + \omega_F \text{Focal}_l + \omega_T \text{Topo}_l + \alpha_u \omega_T \text{Topo}_u + \omega_C \text{Contr}_{l,u} \qquad (3)$$

In Equation (3), Dice, Focal, Topo, and Contr represent the dice loss, focal loss, topology loss, and contrastive loss, respectively. The subscript $u$ denotes that the loss is applied to unlabeled data, while the subscript $l$ denotes that the loss is applied to labeled data. Note that contrastive loss is applied on the combined batch of labeled and unlabeled data. $\omega_D$, $\omega_F$, $\omega_T$, $\omega_C$ are the weights for each loss term. $\alpha_u$ is a factor used to account for the ratio of unlabeled and labeled data in the batch during training.

To effectively train a diverse task that requires aggregating multiple loss functions, a dynamic weighting mechanism is implemented during training. Figure 2 shows the weighting of the losses throughout the training. In the initial stages of training, the contrastive and segmentation losses are given higher weight as the visual representation and segmentation need to be optimized first as they are crucial for the classification and topology tasks. As the training progresses, the focus shifts to optimizing the topological properties of the segmentation output. Once the segmentation is well-optimized, the model can be trained on the classification task more effectively.

## 3. Experiments and Results

### 3.1. Dataset

#### 3.1.1. NTUH

We collected 7,378 portable supine CXRs from the National Taiwan University Hospital (NTUH) which are labeled with the misplacement state of NGT and ETT, as well as segmentation annotation for the catheters and relevant anatomical landmarks. This dataset includes a training set and a testing set. The training set (n=5,767) was collected from the NTUH PACS database through keyword search of the radiology information system, and it includes data collected from the emergency department (ED) or intensive care units between 2015 and 2019. The testing set (n=1,611) was randomly sampled from the ED and is a mixture of data from 2020 from NTUH and data from 2015-2020 from the NTUH Yunlin Branch to test the chronological and geographical generalizability of the model. The testing set has limited amount of abnormal data as it reflects real-world prevalence. Each CXR is

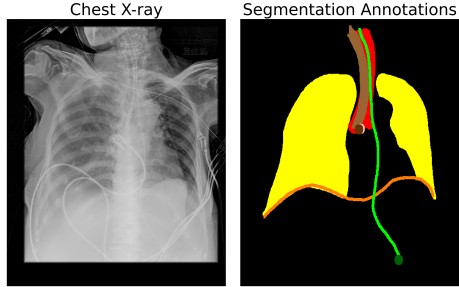

Figure 3: Sample of CXR and its segmentation annotations, including ETT (light brown), NGT (light green), Lung (yellow), Trachea (red), Diaphragm (orange), ETT tip (dark brown), NGT tip (dark green) and Carina (pink).

labeled as Abnormal (incorrect placement), Normal (correct placement), or NA (catheter absence) for each catheter. The distribution of different classes is shown in Appendix B.

Figure 3 is a sample of CXR and its segmentation annotations. The segmentation annotation includes labels for ETT, NGT, Lung, Trachea, Diaphragm, as well as three additional channels for ETT tip, NGT tip and Carina, that was extracted from the segmentation mask of ETT, NGT, and Trachea.

### 3.1.2. CLiP

This Catheter and Line Position (CLiP) dataset is provided by (Tang et al., 2021). It contains 30,083 CXRs that includes data with misplaced catheter. It contains misplacement label of NGT and ETT, as well as their segmentation mask. In our work, this dataset is used as unlabeled data for semi-supervised learning.

### 3.2. Training

In the experiment, the data is split into training and validation sets with a ratio of 4 : 1 using a stratified sampling method, ensuring the same proportion of each class. Images with catheter malposition were oversampled during training to balance the number across each class.

As preprocessing, each input image is resized to $512 \times 512$ before applying contrast-limited adaptive histogram equalization (CLAHE) (Pizer et al., 1990) to enhance the image. Nine different parameters pairs were used for the CLAHE, by taking 0.02, 0.05, 0.1 of the image size as the kernel size and 0.02, 0.05, 0.1 as the clip limit. The image is then processed with CLAHE nine times and the results are stacked into a nine-channel array to improve the visibility.

During the training process, the batch size was 36, where labeled and unlabeled data were randomly sampled with ratio 3:1. The AdamW optimizer (Loshchilov and Hutter, 2017) was used, and the learning rate was 3e-4, scheduled using Cosine Annealing with Warm Restarts (Loshchilov and Hutter, 2016). To have more stable and realistic results of the segmentation task, the output and target segmentation masks were applied with dilation

Table 1: Segmentation Dice score and classification AUC of each method.

| Previous Studies | Dataset (n=size) | Dice (95% CI) | AUC (95% CI) |
|---|---|---|---|
| FS (Sullivan et al., 2020) | Custom pediatric CXR (n=1,390) | 0.74 | - |
| FS (Elaanba et al., 2021) | CLiP (n=30,083) | - | 0.80 |
| FS (Lakhani, 2017) | Custom CXR with ETT (n=300) | - | 0.81 |
| FS (Singh et al., 2019) | Custom CXR with NGT (n=5,754) | - | 0.87 (0.80, 0.94) |
| SD (Gherardini et al., 2020) | Custom fluoroscopy (n=12,207) | 0.55 | - |
| SD (Aryal and Yahyasoltani, 2021) | CLiP (n=30,083) | - | 0.96 |
| Ours | | | |
| FS (baseline) | NTUH (n=7,378) | 0.517 (0.512, 0.522) | **0.979** (0.970, 0.987) |
| TL | NTUH + CLiP (n=37,461) | 0.385 (0.379, 0.390) | 0.974 (0.964, 0.983) |
| CL | NTUH + CLiP (n=37,461) | 0.584* (0.579, 0.590) | 0.968 (0.955, 0.979) |
| CL + TL | NTUH + CLiP (n=37,461) | **0.614*** (0.609, 0.619) | 0.977 (0.968, 0.986) |

FS = Fully-supervised, SD = Synthetic data

TL = Semi-supervised with topology loss, CL = Semi-supervised with contrastive loss

Asterisk (*) denotes statistical significance over baseline with $p$-value $< 0.05$

**Bold** denotes the best performance results

using a $3 \times 3$ square kernel. Otherwise, the line-like shape of the segmentation could cause the loss and metrics to be oversensitive.

The segmentation loss used in the experiment is the spatial-weighted dice loss, which is a modification of the dice loss (Milletari et al., 2016) that assigns more weight to pixels closer to the point of interests (POIs), namely NGT tip for NGT, ETT tip for ETT, and Carina for Trachea. The classification loss used is focal loss (Lin et al., 2017), with $\gamma = 2$ and $\alpha = 3$ for misplaced catheter. This loss also includes label smoothing (Müller et al., 2019) of 0.1.

Our experiments showed that the best performance was achieved with these hyperparameters and this ratio of labeled and unlabeled data. These should be fine tuned depending on the dataset. To see the stability and overfitting issue of the training, see training curves in Appendix C.

### 3.3. Results

The performance of our proposed method is evaluated on the NTUH testing set, using metrics including Dice score and area under the receiver operating characteristic curve (AUC). We also compute 95% confidence intervals and p-values over baseline using bootstrapping (with n=1000) and Wilcoxon signed-rank test. Dice score is used to measure the agreement between the predicted and target segmentation masks. The predicted mask is thresholded at 0.5, and the result is an average over all predicted classes. AUC is used to evaluate the classification performance. It is computed with one-vs-rest for NA, Normal, and Abnormal and then take the macro average. The result is the mean AUC over all catheters. The results are shown in Table 1. For detailed results of individual classes, see Appendix D. For result visualizations, see Appendix E.

## 4. Discussion

Previous studies on catheter segmentation have used supervised learning (Sullivan et al., 2020), and have achieved a Dice score of 0.74, and for catheter misplacement prediction, AUC above 0.8 (Elaanba et al., 2021; Lakhani, 2017; Singh et al., 2019). However, these

studies have been limited by the availability of labeled data. To address this problem, some studies have used synthetic data methods, such as imposing predefined segmentation labels on X-ray images (Gherardini et al., 2020) or using generative adversarial networks (GANs) to synthesize catheters in X-ray images (Aryal and Yahyasoltani, 2021), achieving Dice score of 0.55 and AUC of 0.96. It is important to note that these mentioned studies, although similar in nature, have slightly different tasks and datasets. Therefore, direct comparison between them is not possible, and they are only listed for reference in Table 1.

In contrast, our proposed method utilizes a semi-supervised learning approach by using a contrastive loss to train image representations and a topology loss to train segmentation with topological priors. This allows the model to learn from labeled and unlabeled data. This approach improves the baseline performance of the segmentation significantly and reaches a Dice score of 0.614 while maintaining a performance similar to the fully-supervised baseline for misplacement prediction at AUC 0.977. The effect of using only topology loss and contrastive loss on classification performance is limited, as no additional information about the classification labels is provided during the semi-supervised learning process.

From the results shown in Table 1, it is clear that using the topology loss alone decreases the Dice score from 0.517 to 0.385. However, when combined with contrastive loss, the Dice score improves to 0.584, which is higher than using contrastive loss alone. This can be attributed to the fact that the topology loss is only effective when the initial segmentation results are already reasonable. On the other hand, using contrastive loss allows the model to learn visual representations from unlabeled data, thus improving the Dice score from 0.517 to 0.584. Therefore, by adding topology loss on top of the contrastive loss, the model can improve the initial segmentation performance from the contrastive loss and then apply the topology loss to restrict the number of connected components more effectively, resulting in an improved Dice score of 0.614. This result is statistically significant over the baseline with a p-value of less than 0.05.

## 5. Conclusion

In summary, this study proposed a semi-supervised learning method that combines contrastive loss and topology loss to improve the performance of catheter segmentation and misplacement prediction on CXRs. The method utilizes a U-Net architecture with a classification head and learns from both labeled and unlabeled data by using a contrastive loss to train image representations and a topology loss to train segmentation with topological priors. The results of the study show that the proposed method improves the baseline performance of segmentation significantly while maintaining a robust classification accuracy. Our proposed method of combining contrastive and topology loss for semi-supervised learning could be extended to other tasks to improve segmentation performance if additional unlabeled data is available and a topological prior is known.

## Acknowledgments

This study was supported by the National Science and Technology Council, Taiwan (NSTC 110-2115-M-002-008-MY3, NSTC 112-2114-M-002-004-).

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

## Appendix A. Model Architecture

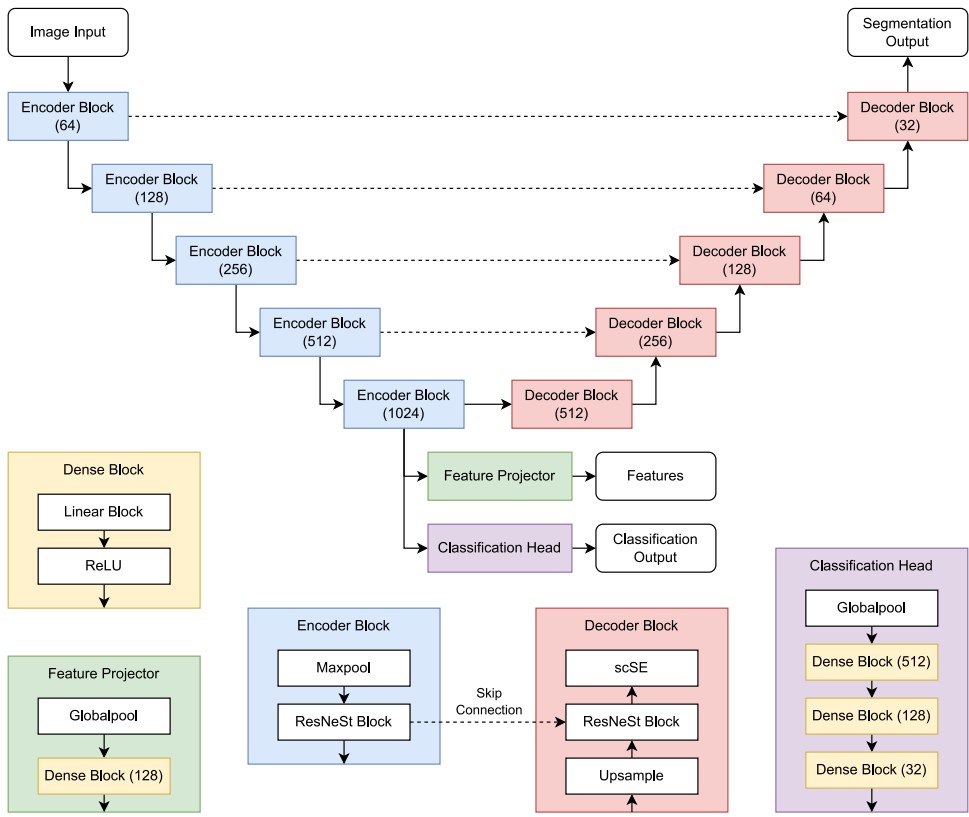

Figure 4: U-Net with feature projector and classification head.

## Appendix B. Data Distributions

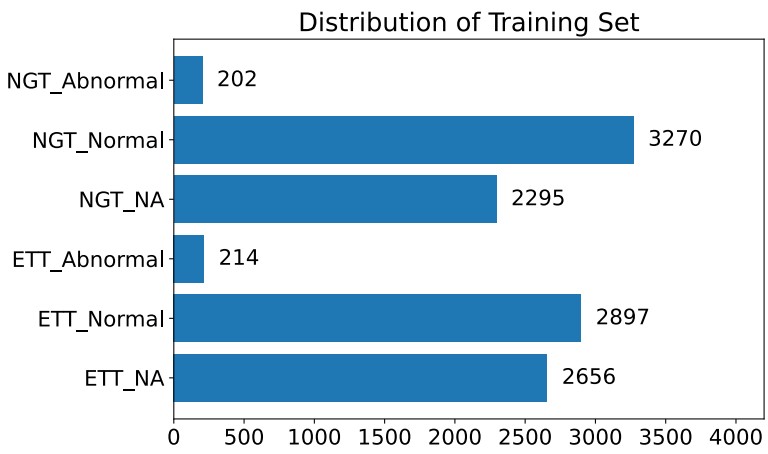

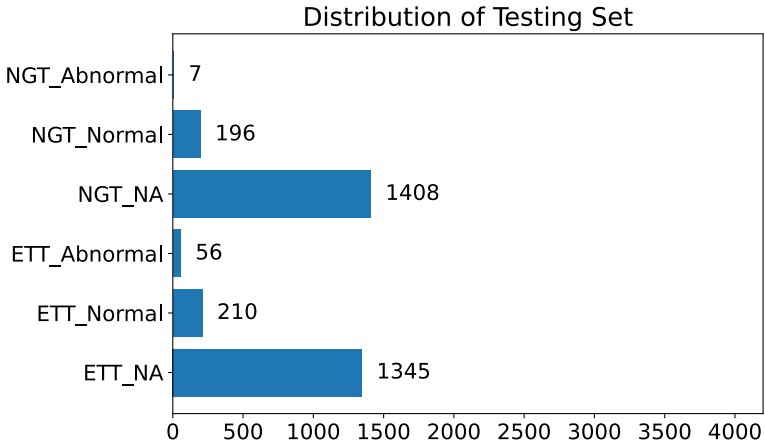

Figure 5: The distribution of catheter misplacement of the training set and testing set in NTUH dataset.

## Appendix C. Training Curves

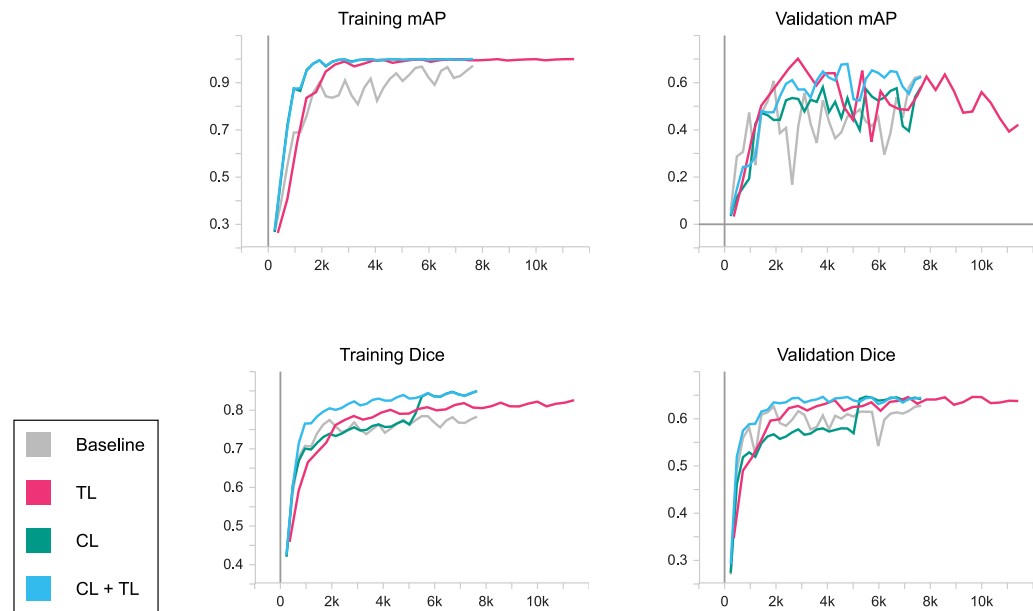

Figure 6: The training and validation mean average precision (mAP) of detecting catheter malposition, and the average Dice score, throughout the training of each method. The y-axis is the metric, and the x-axis is the training steps. TL = Semi-supervised with Topology loss, CL = Semi-supervised with Contrastive loss

# Appendix D. Detailed Results

Table 2: Segmentation Dice score of each class of each method.

| Class | Baseline | TL | CL | CL + TL |
|---|---|---|---|---|
| ETT | 0.301 (0.278, 0.324) | 0.162 (0.146, 0.181) | 0.410 (0.386, 0.434) | **0.669**\* (0.647, 0.692) |
| Trachea | **0.768** (0.763, 0.773) | 0.740 (0.734, 0.745) | 0.758 (0.753, 0.764) | 0.757 (0.752, 0.763) |
| NGT | 0.094 (0.081, 0.105) | 0.092 (0.080, 0.103) | **0.232** (0.213, 0.253) | 0.147\* (0.131, 0.162) |
| Lung | 0.903 (0.901, 0.906) | 0.902 (0.899, 0.904) | 0.918 (0.916, 0.921) | **0.923**\* (0.921, 0.925) |
| Diaphragm | 0.508 (0.502, 0.515) | 0.514 (0.507, 0.520) | 0.525 (0.518, 0.532) | **0.534**\* (0.527, 0.541) |
| ETT tip | 0.749 (0.730, 0.769) | 0.232 (0.214, 0.250) | 0.816 (0.801, 0.833) | **0.857**\* (0.842, 0.871) |
| NGT tip | 0.580 (0.558, 0.603) | 0.224 (0.205, 0.242) | **0.841** (0.825, 0.858) | 0.826\* (0.809, 0.844) |
| Carina | **0.230** (0.219, 0.242) | 0.212 (0.201, 0.223) | 0.174 (0.163, 0.187) | 0.196 (0.184, 0.210) |

TL = Semi-supervised with topology loss, CL = Semi-supervised with contrastive loss
Asterisk (\*) denotes statistical significance over baseline with $p$-value $< 0.05$
**Bold** denotes the best performance results

Table 3: Classification AUC of each class of each method.

| Class | Baseline | TL | CL | CL + TL |
|---|---|---|---|---|
| ETT | 0.963 (0.944, 0.979) | 0.965 (0.950, 0.977) | 0.947 (0.925, 0.966) | **0.966**\* (0.953, 0.979) |
| NGT | **0.995** (0.992, 0.998) | 0.984 (0.971, 0.996) | 0.989 (0.974, 0.998) | 0.989 (0.975, 0.999) |

TL = Semi-supervised with topology loss, CL = Semi-supervised with contrastive loss
Asterisk (\*) denotes statistical significance over baseline with $p$-value $< 0.05$
**Bold** denotes the best performance results

## Appendix E. Result Visualizations

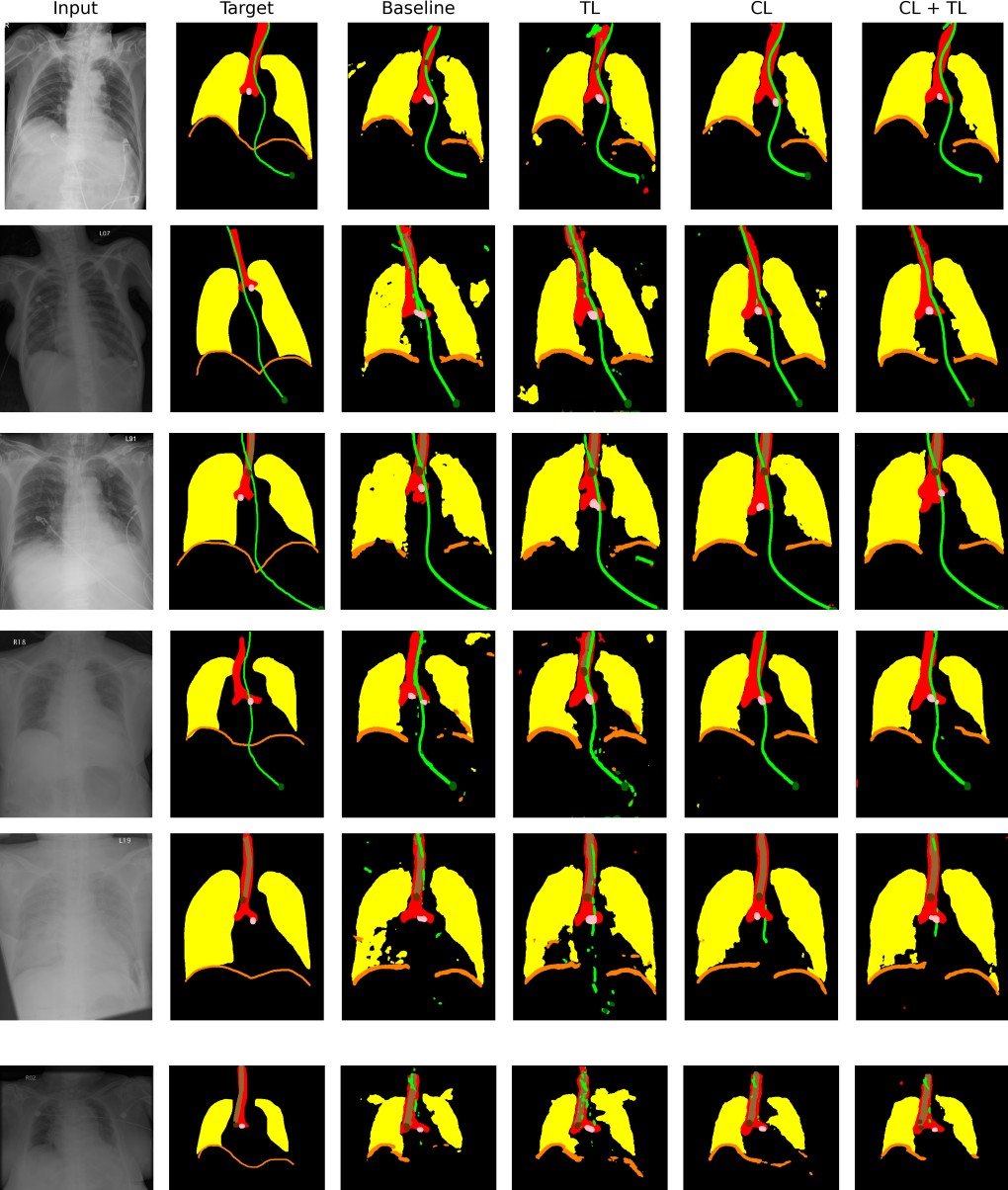

Figure 7: Result visualizations for each method. From left to right are input CXRs, segmentation targets, and sample predictions from each method. From top to bottom are six different samples. TL = Semi-supervised with topology loss, CL = Semi-supervised with contrastive loss.

