# OpenReview forum: "Semi-supervised Learning with Contrastive and Topology Losses for Catheter Segmentation and Misplacement Prediction"
_MIDL.io/2023/Conference — MIDL 2023 Poster_

### Official Review · Reviewer_4RL8 · 2023-02-04

**Confidence:** 5
**Preliminary Rating:** 3

**Summary:**

--Author proposed a novel semi-supervised method for learning classification and segmentation tasks jointly.

--The Contrastive Loss and Topology Loss are used for improving the results.

--The proposed method is evaluated on an in-house dataset (it shown as ***** in paper for anonymize) with improved performance.

**Strengths:**

--The author proposed a semi-supervised learning method that combines contrastive loss and topology loss to improve the performance of catheter segmentation and misplacement prediction on chest X-ray images.

--the details of performance is presented in Table.1, especially for the ablation studies of different losses.

--the application is valuable.

--compared to the baseline method, the proposed method could produce better segmentations.

--compared to other methods (except the baseline method), the proposed method achieve higher AUC score.

**Weaknesses:**

major:

--semi-supervised is well studied, no comparison with other semi-supervised methods

minor:

--The classification results are similar for all proposed methods and baseline method.

--it is not clear for me about the Dice results. It is confusing since the multi-class label is obtained from dataset, but only single Dice is represented as averaged. It would be better to show results of multiple classes.

--other metrics could be used for Dice analysis, such as average symmetric surface distance and Hausdorff distance.

--For figure 4, please remove the words on the top right corner of CT image if possible and make clear notation of each label each on image or at caption.

**Deanonymize Review:**

no

**Detailed Comments:**

please see the weaknesses

**Paper Type:**

validation/application paper

**Questions To Address In The Rebuttal:**

--please explain or address the dice analysis which is shown as the minor weakness.

--In Table.1, the results of  Sullivan et al. 2020 (0.74) is confusing with reading the submitted paper alone or without check the paper proposed by Sullivan et al.. Please explain it in more details.

--I didn't find the explanation of dynamic loss weight, please simply explain how those weights are defined.

--Have you tried to use different number of unlabeled data during the training?


--------------------------------------------------------------------------------------------------------------------------------------------------------------------


--please explain other questions which is listed in weakness, if possible but not required

---

### Official Review · Reviewer_AyWb · 2023-02-06

**Confidence:** 4
**Preliminary Rating:** 3

**Summary:**

This paper proposed a novel semi-supervised learning method that combines contrastive loss and topology loss for catheter segmentation and misplacement prediction. The motivation for semi-supervised learning is sufficient, but the novelty of the solution needs to be discovered. In addition, the combination of multiple loss functions makes it more difficult for network training to converge.

**Strengths:**

1) This manuscript developed an algorithm that simultaneously performs catheter segmentation and misplacement prediction using a U-Net architecture with an additional classification head.
2) This manuscript modified topology loss to constraint on a range of topological features.
3) It has to be said that collecting and preprocessing this dataset was a difficult process.


**Weaknesses:**

1) There are few innovations in U-net structure, only one more classification head.
2) Although topology loss has been modified for clinical problems, Contrastive Loss has not. Correspondingly, the title of this article is semi-supervised learning, and the topology loss has limited improvement from the ablation experiment.
3) From the graph on the right side of Fig. 2, the weight of the loss function does not seem to have an extreme point, which makes it a little far-fetched to convince researchers to use your combined loss function.
4) There seems to be too little abnormal data for the test set, especially NGT_abnormal.

**Deanonymize Review:**

no

**Detailed Comments:**

1) The overall architecture seems to be the Fig.2 not the Fig.1, which is shown the combined loss.
2) There are few innovations in U-net structure, only one more classification head.
3) Although topology loss has been modified for clinical problems, Contrastive Loss has not. Correspondingly, the title of this article is semi-supervised learning, and the topology loss has limited improvement from the ablation experiment.
4) From the graph on the right side of Fig. 2, the weight of the loss function does not seem to have an extreme point, which makes it a little far-fetched to convince researchers to use your combined loss function.
5) There seems to be too little abnormal data for the test set, especially NGT_abnormal.

**Paper Type:**

methodological development

**Questions To Address In The Rebuttal:**

1) The overall architecture seems to be the Fig.2 not the Fig.1, which is shown the combined loss.
2) There are few innovations in U-net structure, only one more classification head.
3) Although topology loss has been modified for clinical problems, Contrastive Loss has not. Correspondingly, the title of this article is semi-supervised learning, and the topology loss has limited improvement from the ablation experiment.
4) From the graph on the right side of Fig. 2, the weight of the loss function does not seem to have an extreme point, which makes it a little far-fetched to convince researchers to use your combined loss function.
5) There seems to be too little abnormal data for the test set, especially NGT_abnormal.

---

### Official Review · Reviewer_1EhC · 2023-02-06

**Confidence:** 4
**Preliminary Rating:** 3

**Summary:**

This paper is about catheter segmentation, as a way to improve misplacement prediction (classification task), both in accuracy and interpretability.

The method presented relies on two somewhat recent developments: contrastive learning [1] and topology loss [2]. These two losses enable the use of un-baleled images, creating a semi-supervised setting overall. The contrastive learning part is used to ensure different images have dissimilar embeddings, while the same image with different augmentation gets the same embeddings. The topology loss uses the known prior that the cathether is a single connected component.

Overall, the method is interesting and well motivated, but remains high in heuristics (I counted 13, on top of the typical hyper-parameters found in deep learning), while the evaluation is unclear and unconvincing (see detailed comments and questions to answer during rebuttal). Note that this is addressable during rebuttal, so I invite the authors to engage in the discussion **before the end of the rebuttal period**.

---
[1] Chen, Ting, et al. "A simple framework for contrastive learning of visual representations." International conference on machine learning. PMLR, 2020.
[2] Clough, James R., et al. "A topological loss function for deep-learning based image segmentation using persistent homology." IEEE Transactions on Pattern Analysis and Machine Intelligence 44.12 (2020): 8766-8778.

**Strengths:**

- Well motivated topic and task
- The topology loss is a very good fit for the prior at hand (a single catheter)
- Considerations for data efficiency
- The authors obtain a very good AUC, and pretty good DSC (especially given the narrowness of the object, for which the dice metric can be very sensitive)

**Weaknesses:**

- The paper is at times difficult to follow, especially in Section 2.3 about the topology loss. This is compounded by the fact that the original paper [2] is also short on explanations (especially around the "bar codes" and "birth death" thresholds), so in the end the reviewer (me) is in a difficult position to understand (and henceforth evaluate) the work
- The results are not very well reported: is in unclear what methods are tested on which datasets, with which subset or annotation percentage.
- The data splits are bit unclear: the authors report a training set of 5767 images, and a testing set of 1611 images, but no validation set?

**Deanonymize Review:**

no

**Detailed Comments:**

Table 1 should indicate which dataset is used for each methods, and ideally, test all compared methods on your private dataset. At the moment all I see are unrelated numbers, that I cannot compare between each others.

> It is important to note that these mentioned studies, although similar in nature, have slightly different tasks and datasets. Therefore, direct comparison between them is not possible and they are only listed for reference.

Providing _at least_ the datasets size would help.

The task is a good fit for semi-supervised settings, but yet the authors stick to a 75/25 labeld/unlabeled ratio. Why is that? Why not going higher for the unlabeled set?

Misc:
- Figure 1 is not very useful, by now everyone has a good idea of what UNet is. I would gain some space here to focus on something else, and leave those details to an attached/public codebase.
- Figure 5 should explicit say where the predictions come from (training or testing).
- The writing could be improved at times, for instance, page 3 is hard to understand:
> every image is performed augmentations twice to obtain $B'$


And to rebound on that: why not simply having $B=\\{x_1, ..., x_b\\}$ as original set, and then $B'=\\{x'_1,...,x'_b\\}$ as augmented set? This way $\sigma(i)$ would not be required. (BTW, I think you have a typo when copying from [1], it is a sum to $2b$ and not $2N$.) Per your own definition of $B$, there is an issue with $z_i = g(E(x_i))$ when $i=2b$, as $B$ has elements only up to $x_b$. The contrastive loss would then be?:

$ \\mathcal L_\\text{contrast}(B, B') = - \\frac{1}{b} \\sum_{i=1}^b \\frac{ \\exp(\\text{sim}(z_i, z'_i))/ \\tau }{  \\sum _ {j \\neq i} \\exp(\\text{sim}(z_i, z'_j))/ \\tau  } $

As an aside, [1] seems to basically maximize the similarity between $z_i$ and $z'_i$ while minimizing the similarity between $z_i$ and $z_j \forall j \neq i$, but in a very convoluted way. While it might have been optimal in a multi-class, natural computer vision application, it might not be the case for the task at hand, so investigating alternative similarity metrics and maximization/minimization methods might be worthwhile. This isn't a criticism of your work (it is perfectly valid to build from other works), but I still think that your application could benefit from exploring alternatives.

**Paper Type:**

validation/application paper

**Questions To Address In The Rebuttal:**

### On the intro and motivation
- I am curious, why is it difficult to assess misplacement "manually"?
- What would be the required DSC or AUC for clinical use?
- When reporting the metrics, why sticking to 3 significant numbers, and not 2 or 4?
- Given the prior knowledge, could it be doable to compute common biomarkers on the predicted segmentation to detect failed segmentation automatically? I have in mind the volume, length (from total variation methods), width and number of connected compoments.


### On the method
- While the authors motivate the interpretability of their work by also predicting a segmentation, the classification head is plugged on the network embeddings (bottom of the U-Net). Why is that? Given the motivation I would have expected it to re-use the segmentation map first, with possibly some more (up-scaled) feature maps.
- Given the narrowness of the catheter, doesn't the downscaling to 16x16 resolution, for the topology computation, risks to backfire? (Fusing pixels, muddying the topology)
- How expensive is the computation of the Betti numbers, really?
- I did run out of time while trying to make sense of [2] (I might manage to in the coming week), but from what I understood, in your case, $\beta_0 = \\{0, 1 \\}$. Hence, we would have $\beta_{0,a} = 0$ and $\beta_{0,b}=1$ (basically, at most one bar for $\beta_0$ of length 1)? Doesn't make the first part of Eq. (2) to be always ignored? It is entirely plausible that I completely misunderstood, so please correct me during the rebuttal.
- if generalizing to 3D, I would assume that $\beta_1$ and $\beta_2$ could be supervised to be 0?

### On the results
- The "fully supervised (baseline)" seems to have lower results than the final model of 75 % annotated 25% unannotated, which is surprising to me. Is this baseline trained on 100% of images, with annotations, or only 75 % of images, with annotations
- The task is a good fit for semi-supervised settings, but yet the authors stick to a 75/25 labeld/unlabeled ratio. Why is that? Why not going higher for the unlabeled set?
- I mentioned earlier that dice is a sensitive metric for such object. Did you consider reporting other metrics for the segmentation?
- Is training stable? Is it affected by over-fitting?

> From the results shown in Table 1, it is clear that using the topology loss alone decreases the performance of Dice score from 0.520 to 0.372. However, when combined with contrastive loss, the performance improves to 0.589, which is higher than using contrastive loss alone. This can be attributed to the fact that topology loss is only effective when the initial segmentation results are already reasonable.

Wouldn't it motivate a simple, two stage training? (First only labeled images, then labeled + unlabeled.)

---
**Final rating:** After rebuttal, and responses from the authors, I decided to upgrade my rating from `weak reject` to `borderline`. The authors do not seem to have uploaded a new manuscript, so it is hard to assess, but I am happy with the response and clarification of clinical context provided during rebuttal ; I hope it makes it into the manuscript.

---

### Official Review · Reviewer_ruKP · 2023-02-06

**Confidence:** 3
**Preliminary Rating:** 3
**Recommendation:** Poster

**Summary:**

This paper proposed a semi-supervised learning method that combines a contrastive loss with a topology loss to improve the performance of catheter placement classification and segmentation in chest X-ray images. The prediction model was realized as U-Net with additional classification branch and feature projector. The proposed method improves the segmentation performance significantly compared to the baseline method.

**Strengths:**

This paper formulated the problem catheter segmentation and misplacement, and proposed semi-supervised learning framework to benefit a large amount of unlabeled dataset. The proposed contrastive loss and topology loss sound reasonable and the authors adopted implementation techniques to leverage the segmentation/classification performance for the imbalanced data.

**Weaknesses:**

The results section seems to be too short and needs improved. Dice coefficients are averaged for all ROIs. Since the volumes of each ROI are quite different, separate DSC results for each ROI are needed. Classification performance can be shown as a confusion matrix as it has multiple classes instead of binary classification.

**Deanonymize Review:**

no

**Detailed Comments:**

The ratio of labeled vs unlabeled data was fixed as 3:1 in this paper. It would be interesting to see the different performance results as the ratio varies. Did the performance reach the plateau and decrease when the number of unlabeled data exceeds this ratio?

**Paper Type:**

methodological development

**Questions To Address In The Rebuttal:**

-Topology loss was computationally expensive and implemented on CPU. Does this mean that the whole training was done on CPU, or only the computation of topology loss was done on CPU? Please clarify. I’m also concerned about poor resolution of 16x16 after downsampling. Does this image still have enough anatomical details to compute topology loss?
-In table 1, the statistical significance was tested compared to the baseline. Was the improvement of CL+TL over CL statistically significant?

---

### Official Review · Reviewer_TRgP · 2023-02-06

**Confidence:** 4
**Preliminary Rating:** 5
**Recommendation:** Oral

**Summary:**

The paper presents a semi-supervised learning approach for detecting and predicting catheter misplacement in chest X-ray images. Considering the known topology of catheters, the method uses a combined loss consisting of contrastive and topology losses to train a U-Net architecture with an added classification head to simultaneously perform catheter segmentation and misplacement prediction. The results show improved performance in catheter segmentation and misplacement prediction. This approach can be extended to other tasks, given the availability of unlabeled data and known topological information.

**Strengths:**

The paper is well-written and well-structured. It clearly and concisely explains the problem and challenges associated with catheter placement evaluation in medical practice using chest X-rays, as well as the proposed solution. The authors comprehensively state the issues with obtaining labeled datasets for training computer-aided detection models and the class imbalance in such datasets. They have done an excellent job of demonstrating the effectiveness of their proposed method and its potential for real-world applications. This paper provides valuable insights into computer-aided detection for catheter placement evaluation and is a valuable resource for future research.


**Weaknesses:**

The baseline is not clearly defined and it is unclear why a Fully-supervised approach (Sullivan et al., 2020) with a dice score of 0.74 is not considered as the baseline instead of another Fully-supervised baseline with a dice score of 0.520. The authors should provide a clear explanation of the baseline to ensure a better understanding of the results.

**Deanonymize Review:**

no

**Paper Type:**

both

**Questions To Address In The Rebuttal:**

The paper provides a clear overview of the problem and the proposed solution. The method is explained in detail and the proposed semi-supervised approach using a combination of contrastive and topology losses to train a U-Net architecture shows improved results in catheter segmentation and misplacement prediction. However, the definition of the baseline is not clear. It is not evident why the Fully-supervised approach (Sullivan et al., 2020) with a dice score of 0.74 is not considered as the baseline instead of another Fully-supervised baseline with a dice score of 0.520. The authors should clarify the definition of the baseline for better understanding of the results. Additionally, increasing the text size in the figures would improve the visibility of the data, making it easier to interpret.

---

### Meta-Review · Area_Chair_enQV · 2023-02-23

**Recommendation:** Accept (Poster)
**Confidence:** 4

**Metareview:**

This work presents a semi-supervised learning approach for detecting and predicting catheter misplacement in chest X-ray images. The paper clearly explains the challenges in this work, and shows the semi-supervised solution of solving these challenges. The proposed method is useful in clinical diagnosis due to its ability of utilizing unlabeled data.

Pros:
- The idea of imposing domain knowledge by a topology loss is novel.
- The semi-supervised approach can be valuable in clinical applications

Cons:
- The ratio between labeled and unlabeled data (3:1) could be too high in real applications
- The comparison with other semi-supervised methods can be included.

Among five reviewers, one rated strong accept, other four rated borderline. After balancing between pros and cons, I recommend to accept this paper.